# Bone mineral density among virologically suppressed Asians older than 50 years old living with and without HIV: A cross-sectional study

**Lalita Wattanachanya**[1,2], **Sarat Sunthornyothin**[1,2], **Tanakorn Apornpong**[3], **Hay Mar Su Lwin**[3], **Stephen Kerr**[3,4,5], **Sivaporn Gatechompol**[3,6], **Win Min Han**[3,6], **Thanathip Wichiansan**[3], **Sarawut Siwamongsatham**[7], **Pairoj Chattranukulchai**[8], **Tawatchai Chaiwatanarat**[9], **Anchalee Avihingsanon**[3,6]*, **HIV-NAT 207/006 study team**[¶]

1 Division of Endocrinology and Metabolism, Department of Medicine, Faculty of Medicine, Chulalongkorn University, Bangkok, Thailand, **2** Excellence Center for Diabetes, Hormone, and Metabolism, King Chulalongkorn Memorial Hospital, Bangkok, Thailand, **3** HIV Netherlands Australia Thailand Research Collaboration (HIV-NAT), Thai Red Cross AIDS Research Centre, Bangkok, Thailand, **4** Biostatistics Excellence Centre, Faculty of Medicine, Chulalongkorn University, Bangkok, Thailand, **5** The Kirby Institute, University of New South Wales, Sydney, NSW, Australia, **6** Center of Excellence in Tuberculosis, Faculty of Medicine, Chulalongkorn University, Bangkok, Thailand, **7** Division of Ambulatory and Hospital Medicine, Faculty of Medicine, Chulalongkorn University, Bangkok, Thailand, **8** Division of Cardiovascular Medicine, Faculty of Medicine, Chulalongkorn University, King Chulalongkorn Memorial Hospital, Bangkok, Thailand, **9** Department of Radiology, Faculty of Medicine, Chulalongkorn University, Bangkok, Thailand

¶ The team members of HIV-NAT 207/006 study team can be found in the Acknowledgments.
* anchaleea2009@gmail.com

**Data Availability Statement:** Data cannot be shared publicly because of confidentiality concerns. Data are available from the

## Abstract

There are limited data regarding bone health in older people living with HIV (PWH), especially those of Asian ethnicity. We aimed to determine whether BMD in well-suppressed HIV-infected men and women aged $\geq$ 50 years are different from HIV-uninfected controls. In a cross-sectional study, BMD by dual-energy X-ray absorptiometry and calciotropic hormones were measured. A total of 481 participants were consecutively enrolled (209 HIV+ men, 88 HIV- men, 126 HIV+ women and 58 HIV- women). PWH were on average 2.5 years younger [men: 55.0 vs. 57.5 yr; women: 54.0 vs. 58.0 yr] and had lower body mass index (BMI) [men: 23.2 vs. 25.1 kg/m$^2$; women: 23.1 vs. 24.7 kg/m$^2$] compared to the controls. The median duration since HIV diagnosis was 19 (IQR 15–21) years in men and 18 (IQR 15–21) years in women. Three-quarters of PWH had been treated with tenofovir disoproxil fumarate-containing antiretroviral therapy for a median time of 7.4 (IQR 4.5–8.9) years in men and 8.2 (IQR 6.1–10) years in women. In an unadjusted model, HIV+men had significantly lower BMD (g/cm$^2$) at the total hip and femoral neck whereas there was a tend toward lower BMD in HIV+women. After adjusting for age, BMI, and other traditional osteoporotic risk factors, BMD of virologically suppressed older PWH did not differ from participants without HIV (P>0.1). PWH had lower serum 25(OH)D levels but this was not correlated with BMD. In conclusion, BMD in well-suppressed PWH is not different from non-HIV people, therefore, effective control of HIV infection and minimization of other traditional osteoporosis

Chulalongkorn University Institutional Review Boards/ Ethics Committee (contact via chavalun. r@hivnat.org) for researchers who meet the criteria for access to confidential data.

**Funding:** The authors received no specific funding for this work.

**Competing interests:** KR received honoraria or consultation fees and/or participation in a company sponsored speaker's bureau from: ViiV, Merck, Jensen-Cilag, Mylan and Gilead. The rest of the authors declare no conflict of interest. This does not alter our adherence to PLOS ONE policies on sharing data and materials

risk factors may help maintain good skeletal health and prevent premature bone loss in Asian PWH.

**Clinical trial registration:** Clinicaltrials.gov # NCT00411983.

## Introduction

Longevity of people living with HIV (PWH) has dramatically improved in the last two decades, with life expectancy approaching that of the general population, predominantly due to expanded access to highly effective antiretroviral therapy (ART) [1]. The population of PWH aged ≥50 years across many world regions is expected to increase steadily to 21% by year 2030 [2]. As more PWH live longer, it is expected that medical comorbidities such as cardiovascular diseases, renal dysfunction, cancer, and bone diseases will increase [3, 4].

Low bone mineral density (BMD) has been widely documented in PWH. Prior meta-analyses indicated that the prevalence of low BMD in adults living with HIV was three times higher compared to people without HIV [5–7]. In addition, some studies suggested that fractures occurred more frequently in PWH [8–10]. Underlying mechanisms leading to reduced bone mass in PWH are believed to be multifactorial, including both traditional and HIV-specific risk factors. PWH are likely to have risk factors related to osteoporosis such as smoking, low body weight, nutritional deficiencies, and hypogonadism [11–15]. Several studies found that vitamin D deficiency, which may adversely affect calcium and skeletal homeostasis, was more prevalent in PWH compared to people without HIV [16–18]. In addition, HIV itself and chronic inflammation/immune activation may negatively affect bone mass by altering osteoblast and/or osteoclast function [19, 20]. Specific antiretroviral drugs such as tenofovir disoproxil fumarate (TDF) may also lead to a reduction of bone mass and increased risk of fracture [21].

Most of the data mentioned above are from studies conducted in western countries. Data regarding bone health in Asians living with HIV, especially in persons over 50 years old, are currently limited. Therefore, we sought to determine whether BMD in PWH over 50 years old are different from people without HIV. This study also compared the levels of calciotropic hormones, including 25-dihydroxyvitamin D (25(OH)D), and bone turnover markers (BTMs) between PWH and people without HIV.

## Methods

### Study population and settings

This cross-sectional study was conducted between January 2016 and June 2017. Three hundred and fifty-eight PWH over 50 years of age who were virologically suppressed (HIV-RNA <50 copies/mL) in a long-term cohort (HIV-NAT 006: clinical trial number NCT00411983) at the HIV Netherlands Australia Thailand Research Collaboration (HIV-NAT), Bangkok, Thailand, were asked to participate in this study. Age (within 10-year age bands) and sex-matched controls were recruited from those visiting the King Chulalongkorn Memorial Hospital for their annual medical checkup during the study period. Participants with known history of metabolic bone diseases, multiple myeloma, cancer, inflammatory bowel disease, currently on glucocorticoids or anticonvulsant agents, and/or currently admitted to the hospital were excluded. These criteria applied to both people living with and without HIV. Three hundred thirty-five out of 358 PWH and 146 healthy controls who met the inclusion criteria were enrolled (Fig 1).

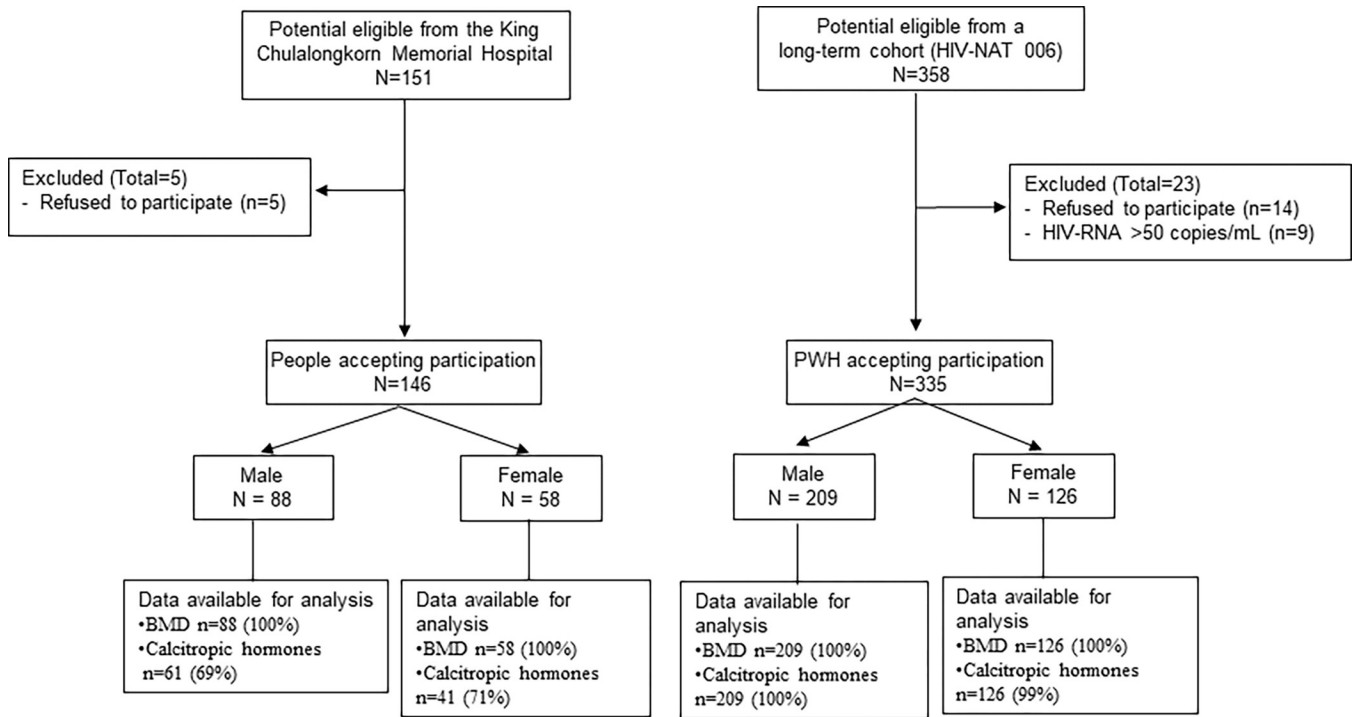

**Fig 1. Participants recruitment and investigation flow diagram.**

Demographic and medical data of all participants were extracted from the clinic's electronic health database and chart review: reproductive history, physical activity, current medications, medical illness, traditional risks for osteoporosis and fracture history. For PWH, the following data were collected: ART history, nadir CD4 cell count, and duration of HIV infection. Whole blood samples were drawn on the day of the clinic visit after at least 8 hours of fasting and plasma was stored at -80°C until assayed. CD4 cell counts, HIV-RNA viral load, creatinine, calciotropic hormones, and BTMs, including calcium, phosphorus, albumin, intact parathyroid hormone (ICMA; intact PTH assays Roche Elecsys®), serum 25(OH)D levels (CLIA; *DiaSorin* LIAISON®), serum procollagen type 1 N-terminal propeptide (P1NP), and C-terminal cross-linking telopeptide of type I collagen (CTX) (CLIA; Roche Diagnostics, Mannheim, Germany) were measured. Inflammatory markers, high sensitivity C-reactive protein (hs-CRP; Roche Diagnostics GmbH, Mannheim, Germany), and Interleukin 6 (IL-6; ECLIA; Roche Diagnostics GmbH, Mannheim, Germany) were also measured. The intra- and inter-assay coefficients of variations for 25(OH)D assay were less than 6%. Vitamin D deficiency and insufficiency were defined as a serum 25(OH)D below 20 ng/mL and within 20–29 ng/mL, respectively [22]. The same tests, with the exception of viral load and CD4 count, were ordered for people without HIV.

BMD at the lumbar spine (L1 to L4), total hip, and femoral neck, as well as the percentage of body fat were measured using dual-energy X-ray absorptiometry on a QDR 4500 bone densitometer (Hologic, Inc., Bedford, MA). BMD T-scores were analyzed using Asian population reference databases, supplied by the manufacturer. Osteoporosis was defined as a T-score of ≤ -2.5 standard deviation (SD) and osteopenia, or low BMD, was defined as a T-score between -1 and -2.5 SD below the young adult mean value, as per World Health Organization (WHO) criteria [23, 24]. A subset of PWH underwent lateral thoraco-lumbar (T-L) X-ray radiographs at the left lateral position centered at L1 level. Radiographic vertebral fracture was diagnosed

using the Genant's semi-quantitative method [25]. Vertebral bodies from T4 to L4 levels were assessed to identify vertebral fracture.

This study was reviewed and approved by the Institutional Review Board of the Faculty of Medicine, Chulalongkorn University, Bangkok, Thailand. HIV-NAT 207 IRB approval No. 442/58 and HIV-NAT 006 study IRB approval No. 589/2007; REC. No. 161/45. Written informed consent was obtained from all of the participants.

## Statistical analysis

Descriptive data were reported as median (IQR), mean (SD), or number (percentage). Formal comparisons of the continuous demographic data and BMD covariates between the study groups were made using Student's t-test or non-parametric equivalents where appropriate; categorical comparisons were made using a Chi-square or Fisher's exact test as appropriate. Univariate analyses were developed for factors associated with BMD, including demographic and anthropometric data, in all participants. In addition to age, sex, alcohol and smoking status, which were well known factors affecting BMD, all variables with p-values ≤ 0.20 were included in the multivariable regression model. Separate models incorporating HIV-disease-related characteristics were developed for PWH for BMD. Collinearity was tested, and nonoverlapping covariates were retained for inclusion in the final models. Subgroup analyses of multivariate linear regression models were developed for serum 25(OH)D and BTMs. Potential determinants with a p-value less than 0.05 were considered statistically significant. Data analysis was performed using Stata version 15.1 (StataCorp LLC, College Station, TX, USA).

## Results

### Characteristics of the study population

From January 2016 to June 2017, a total of 481 patients were consecutively recruited into the study and analysed (209 men living with HIV (HIV+men), 88 men living without HIV (HIV-men), 126 women living with HIV (HIV+women) and 58 women living without HIV: HIV-women). Demographic and clinical characteristics of the study population classified by sex are summarized in Table 1.

Men living with HIV were 2.5 years younger, had lower BMI and percentages of body fat compared to the HIV-men. Regarding factors associated with osteoporosis and/or fracture, HIV+men had higher prevalence of hepatitis B and hepatitis C co-infection. Current smoking, history of low-trauma fracture, family history of osteoporosis/fracture were comparable between the two groups. The median (IQR) duration since HIV diagnosis was 19 (15–21) years with median (IQR) nadir and current CD4 cell counts of 183 (67–256) and 599 (444–791) cells/mm3, respectively.

Likewise, HIV+women were 4 years younger, but they had lower BMI and percentages of body fat than HIV-women. Other risk factors for osteoporosis and/or fractures were similar in both groups. However, HIV+women consumed calcium supplements more than HIV-women. The median (IQR) duration since HIV diagnosis was 18 (15–21) years with median (IQR) nadir and current CD4 cell counts of 174 (120–252) and 685 (527–814) cells/mm$^3$, respectively. The proportion of women who has regular menstruation was higher among HIV+women with shorter years since menopause. Three-quarters of PWH have been treated with TDF-containing ART. Among these, 35% used TDF at half dose. Of note, since this is an observational study, all ART regimens and doses were based on the physician's decision and local guidelines, and the reason behind it may probably because of reduced creatinine clearance or low body weight of the patients.

**Table 1. Demographic and clinical characteristics of the study population stratified by sex.**

| Characteristics | Men (n = 297) | | | Women (n = 184) | | |
|---|---|---|---|---|---|---|
| | HIV- (n = 88) | HIV+ (n = 209) | P-value | HIV- (n = 58) | HIV+ (n = 126) | P-value |
| Age (years) | 57.5 (54.0–62.0) | 55.0 (52.0–60.0) | 0.001 | 58.0 (54.0–62.0) | 54.0 (52.0–59.0) | 0.001 |
| Weight (kg) | 67.6 (60.1–73.8) | 66.0 (57.5–72.3) | 0.12 | 60.6 (53.2–64.8) | 55.1 (51.1–62.0) | 0.004 |
| BMI (kg/m$^2$) | 25.1 (22.4–27.3) | 23.2 (21.0–25.3) | <0.001 | 24.7 (22.2–28.2) | 23.1 (20.8–25.4) | 0.001 |
| Body fat (%) | 22.6 (18.8–25.5) | 19.2 (15.2–23.2) | 0.001 | 34.3 (31.9–36.8) | 31.2 (26.3–35.3) | 0.001 |
| Current residential location (% Bangkok) | 11.7 | 48.9 | <0.001 | 33.3 | 49.5 | 0.12 |
| History of all fractures, n (%) | 15 (17.0) | 44 (21.0) | 0.47 | 7 (12.1) | 18 (14.3) | 0.75 |
| History of fragility fracture, n (%) | 3 (3.4) | 4 (1.9) | 0.42 | 3 (5.2) | 6 (4.8) | 0.86 |
| FH of osteoporosis/fracture, n (%) | 20 (22.7) | 40 (19.1) | 0.48 | 5 (8.6) | 21 (16.7) | 0.15 |
| Current smoking, n (%) | 15 (17.1) | 46 (22.0) | 0.33 | 0 (0) | 2 (1.6) | 1.00 |
| Alcohol use (>1 drink /day), n (%) | 19 (21.6) | 25 (12.0) | 0.033 | 2 (3.5) | 5 (4.0) | 1.00 |
| Reproductive history (women) | | | | | | 0.28 |
| Regular menstruation, n (%) | | | | 7 (12.1) | 26 (20.6) | |
| Bilateral ovaries resection, n (%) | | | | 5 (8.6) | 14 (11.1) | |
| • Year since bilateral ovaries resection | | | | 9 (5–29) | 12 (7–20) | |
| • Menstruation exhausted, n (%) | | | | 46 (79.3) | 89 (68.3) | |
| • Years since menopause | | | | 10 (5–14) | 6 (2–12) | |
| Chronic medical conditions, n (%) | | | | | | |
| • Hypertension | 34 (38.6) | 115 (55.0) | 0.01 | 13 (22.4) | 36 (28.6) | 0.38 |
| • Dyslipidemia | 22 (25.0) | 129 (61.7) | <0.001 | 10 (17.2) | 79 (62.7) | <0.001 |
| • Type 2 diabetes | 12 (13.6) | 47 (22.5) | 0.08 | 10 (17.2) | 13 (10.3) | 0.19 |
| • Cardiovascular diseases | 3 (3.4) | 6 (2.9) | 0.73 | 0 (0) | 1 (0.8) | 1.00 |
| • Others | 2 (2.3) | 5 (2.4) | 1.00 | 3 (5.2) | 2 (1.6) | 0.18 |
| Hepatitis C seropositive, n (%) | 2 (2.3) | 24 (11.5) | 0.012 | 1 (1.8) | 7 (5.6) | 0.44 |
| Hepatitis B seropositive, n (%) | 6 (6.9) | 32 (15.3) | 0.049 | 2 (3.5) | 9 (7.1) | 0.51 |
| Medications, n (%) | | | | | | |
| • Calcium supplements | 4 (4.6) | 12 (5.7) | 0.79 | 14 (24.1) | 11 (8.7) | 0.005 |
| • Multivitamins | 9 (10.2) | 14 (6.7) | 0.30 | 10 (17.2) | 13 (10.3) | 0.19 |
| • Insulin | 0 (0) | 8 (3.8) | 0.11 | 0 (0) | 2 (1.6) | 1.00 |
| • Oral hypoglycemic agents | 8 (9.1) | 40 (19.1) | 0.032 | 9 (15.5) | 13 (10.3) | 0.31 |
| • Statins | 21 (23.9) | 110 (52.6) | <0.001 | 9 (15.5) | 77 (61.1) | <0.001 |
| • Steroids (ever) | 0 (0) | 1 (0.5) | 1.00 | 0 (0) | 0 (0) | N/A |
| • Proton pump inhibitors | 0 (0) | 4 (1.9) | 0.32 | 2 (3.5) | 0 (0) | 0.09 |
| HIV and ART characteristics | | | | | | |
| Presumptive transmission route, n (%) | | | | | | |
| • Men who have sex with men | | 43 (20.6) | | | N/A | |
| • Heterosexuals | | 140 (67.0) | | | 110 (87.3) | |
| • Injecting drugs use | | 2 (0.9) | | | 0 (0) | |
| • Unknown | | 22 (10.5) | | | 14 (11.1) | |
| Years since HIV diagnosis | | 19 (15–21) | | | 18 (15–21) | |
| History of AIDS-defined illness (%) | | 147 (70.3) | | | 59 (46.8) | |
| Nadir CD4 cell count (cells/mm$^3$) | | 183 (67–256) | | | 174 (120–252) | |
| Current CD4 cell count (cells/mm$^3$) | | 599 (444–791) | | | 685 (527–814) | |
| HIV-1 RNA before ARV initiation (log$_{10}$ copies/mL) | | 4.72 (4.22–5.20) | | | 4.61 (4.09–5.12) | |
| Duration of ART (year) | | 16.2 (13.3–19.1) | | | 16.1 (12.6–18.9) | |

*(Continued)*

**Table 1.** (Continued)

| Characteristics | Men (n = 297) | | | Women (n = 184) | | |
|---|---|---|---|---|---|---|
| | HIV- (n = 88) | HIV+ (n = 209) | P-value | HIV- (n = 58) | HIV+ (n = 126) | P-value |
| Current ART, n (%)[a] | | | | | | |
| • NNRTI-based | | 110 (52.6) | | | 78 (61.9) | |
| • PI-based | | 72 (34.5) | | | 36 (28.6) | |
| • PI and NNRTI | | 15 (7.2) | | | 8 (6.4) | |
| • PI- Integrase inhibitor | | 12 (5.7) | | | 4 (3.2) | |
| • EFV | | 58 (27.8) | | | 31 (24.6) | |
| • Non TDF-based | | 59 (27.7) | | | 32 (24.4) | |
| • TDF-based | | 152 (72.7) | | | 94 (74.6) | |
| TDF full dose (300 mg/day) | | 104 (68.4) | | | 58 (61.7) | |
| TDF reduced dose (150 mg/day) | | 48 (31.6) | | | 36 (38.2) | |
| • TDF exposure (year) | | 7.4 (4.5–8.9) | | | 8.2 (6.1–10) | |

Data are reported as median (IQR), mean (SD), or number (percentage). The difference between PWH and people without HIV were compared using Student t-test, Mann-Whitney test, Chi-square test or Fisher's Exact test, where appropriate. HIV-, people living without HIV; HIV+, PWH; BMI, body mass index; FH, family history; ART, antiretroviral therapy; NNRTI, non-nucleoside reverse transcriptase inhibitors; PI, protease Inhibitor; EFV, efavirenz; TDF, tenofovir disoproxil fumarate; N/A, not assessed. [a]Men, n = 173; Women, n = 112.

## Bone mineral density at the lumbar spine, total hip and femoral neck

Unadjusted BMD (grams per square centimeter) of the total hip and femoral neck were significantly lower in HIV+men than HIV- men, while differences among women were not statistically significant (Table 2). No difference in the vertebral BMD was observed in either men or

**Table 2. Unadjusted BMD and T-score at the lumbar spine, total hip and femoral neck.**

| | Men | | | Women | | |
|---|---|---|---|---|---|---|
| | HIV- (n = 88) | HIV+ (n = 209) | P-value | HIV- (n = 58) | HIV+ (n = 126) | P-value |
| **Lumbar spine** | | | | | | |
| BMD (g/cm$^2$) | 0.96 (0.14) | 0.94 (0.16) | 0.11 | 0.88 (0.18) | 0.85 (0.16) | 0.22 |
| T-score, mean (SD) | -0.48 (1.18) | -0.68 (1.35) | 0.09 | -1.5 (3.64) | -1.37 (1.37) | 0.29 |
| T-score, n (%) | | | 0.40 | | | 0.28 |
| >-1.0 | 56 (64.37) | 122 (58.65) | | 28 (49.12) | 45 (36.59) | |
| -2.5 to -1.0 | 29 (33.33) | 74 (35.58) | | 20 (35.09) | 53 (43.09) | |
| ≤-2.5 | 2 (2.30) | 12 (5.77) | | 9 (15.79) | 25 (20.33) | |
| **Total hip** | | | | | | |
| BMD (g/cm$^2$) | 0.91 (0.13) | 0.86 (0.13) | 0.002 | 0.81 (0.17) | 0.79 (0.13) | 0.14 |
| T-score, mean (SD) | -0.21 (0.92) | -0.58 (0.95) | 0.002 | -0.25 (1.15) | -0.53 (1.10) | 0.11 |
| T-score, n (%) | | | 0.047 | | | 1.00 |
| >-1.0 | 68 (77.27) | 135 (64.59) | | 42 (72.41) | 89 (70.63) | |
| -2.5 to -1.0 | 20 (22.73) | 68 (32.54) | | 15 (25.86) | 34 (26.98) | |
| ≤-2.5 | 0 (0.00) | 6 (2.87) | | 1 (1.72) | 3 (2.38) | |
| **Femoral neck** | | | | | | |
| BMD (g/cm$^2$) | 0.74 (0.12) | 0.71 (0.11) | 0.023 | 0.66 (0.21) | 0.64 (0.12) | 0.060 |
| T-score, mean (SD) | -0.86 (0.93) | -1.13 (0.92) | 0.020 | -1.13 (1.18) | -1.48 (1.08) | 0.023 |
| T-score, n (%) | | | 0.016 | | | 0.024 |
| >-1.0 | 49 (55.68) | 80 (38.28) | | 28 (48.28) | 35 (27.78) | |
| -2.5 to -1.0 | 37 (42.05) | 117 (55.98) | | 22 (37.93) | 73 (57.94) | |
| ≤-2.5 | 2 (2.27) | 12 (5.74) | | 8 (13.79) | 18 (14.29) | |

**Table 3. Univariate and multivariate analysis of factors associated with BMD at the lumbar spine, total hip, and femoral neck in men.**

| | Lumbar spine | | | | Total hip | | | | Femoral neck | | | |
|---|---|---|---|---|---|---|---|---|---|---|---|---|
| | Univariate analysis | | Multivariate analysis | | Univariate analysis | | Multivariate analysis | | Univariate analysis | | Multivariate analysis | |
| | Coef (95% CI) | P-value | Coef (95% CI) | P-value | Coef (95%CI) | P-value | Coef (95% CI) | P-value | Coef (95%CI) | P-value | Coef (95% CI) | P-value |
| HIV | -0.02 (-0.06, 0.02) | 0.23 | -0.002 (-0.04, 0.04) | 0.93 | -0.05 (-0.08, -0.02) | 0.002 | -0.03 (-0.06, 0.003) | 0.076 | -0.03 (-0.06, -0.005) | 0.023 | -0.02 (-0.04, 0.01) | 0.26 |
| Age by 10 years | -0.003 (-0.03, 0.03) | 0.82 | 0.0003 (-0.03, 0.03) | 0.99 | -0.02 (-0.04, 0.01) | 0.13 | -0.02 (-0.04, 0.01) | 0.14 | -0.02 (-0.04, -0.001) | 0.038 | -0.02 (-0.04,-0.0004) | 0.045 |
| BMI | 0.01 (0.01, 0.02) | <0.001 | 0.01 (0.01, 0.01) | <0.001 | 0.02 (0.01, 0.02) | <0.001 | 0.01 (0.01, 0.02) | <0.001 | 0.01 (0.01, 0.02) | <0.001 | 0.01 (0.01, 0.01) | <0.001 |
| Current smoking | -0.05 (-0.10, -0.01) | 0.018 | -0.04 (-0.09,-0.0002) | 0.049 | -0.06 (-0.09, -0.02) | 0.002 | -0.04 (-0.08, -0.01) | 0.016 | -0.05 (-0.08, -0.02) | 0.001 | -0.04 (-0.07, -0.01) | 0.007 |
| Alcohol use | 0.01 (-0.04, 0.06) | 0.73 | 0.02 (-0.03, 0.07) | 0.48 | 0.003 (-0.04, 0.04) | 0.89 | 0.01 (-0.03, 0.05) | 0.72 | 0.005 (-0.03, 0.04) | 0.81 | 0.01 (-0.02, 0.05) | 0.53 |
| HCV | -0.04 (-0.10, 0.03) | 0.27 | | | -0.001 (-0.05, 0.05) | 0.97 | | | 0.01 (-0.04, 0.06) | 0.71 | | |
| HBV | 0.01 (-0.04, 0.06) | 0.70 | | | -0.01 (-0.06, 0.03) | 0.61 | | | -0.01 (-0.05, 0.03) | 0.62 | | |
| FH of osteoporosis | -0.002 (-0.05, 0.04) | 0.92 | | | 0.015 (-0.022,0.052) | 0.43 | | | 0.018 (-0.015, 0.05) | 0.29 | | |

*Coef = coefficient, BKK, residential location in Bangkok.

All variables with p-value <0.20 in the univariate analysis together with age, BMI, current smoking and alcohol use status were adjusted in the multivariate model. BMI: body mass index; HCV: hepatitis C; HBV: hepatitis B; FH: family history.

women. According to WHO criteria, both HIV+men and HIV+women had a higher proportion of low BMD (T-score ≤ -1.0) at the femoral neck (men: 62% in HIV+ vs. 44% in HIV-, P = 0.047; women: 72% in HIV+ vs. 52% in HIV-, P = 0.024).

In men, univariate analysis revealed that age was negatively correlated with BMD at the femoral neck, while BMI and current smoking, were negatively correlated with BMD at all sites. These variables remained significant in the multivariate analyses. HIV infection was a risk factor for low BMD at the total hip and femoral neck, but only in the univariate model (Table 3). In women, age and lower BMI were negatively correlated with BMD at all sites in both univariate and multivariate analysis. Still menstruating was positively correlated with BMD at the lumbar spine. Similar to men, HIV status was not correlated with BMD in women (Table 4). As for PWH, the years since HIV diagnosis, duration of antiretroviral exposure, type of ART and the presence of AIDS defining illness were not associated with BMD in both men and women (P>0.2).

## Fractures

The percentage of participants with a history of fractures, either all fractures or low-trauma fractures, was similar between PWH and the controls (Table 1). A lateral thoraco-lumbar (T-L) X-ray radiograph was done in 154 PWH (87 men and 67 women). Of these, six men and one woman had morphometric vertebral fractures. There were no significant differences in baseline characteristics, laboratory testing, and the WHO BMD classifications between PWH with and without T-L X ray radiograph.

## Calciotropic hormones, vitamin D levels, and bone turnover markers

Serum 25(OH)D levels were lower in HIV+men [23.3 (IQR 17.9–28.8) vs 25.5 (IQR 23.1–31.5) ng/mL, P = 0.001] and HIV+women [21.9 (IQR 17.9–26.8) vs 24.2 (IQR 21.5–28.65) ng/mL,

**Table 4. Univariate and multivariate analysis of factors associated with BMD at the lumbar spine, total hip, and femoral neck in women.**

| | Lumbar spine | | | | Total hip | | | | Femoral neck | | | |
|---|---|---|---|---|---|---|---|---|---|---|---|---|
| | Univariate analysis | | Multivariate analysis | | Univariate analysis | | Multivariate analysis | | Univariate analysis | | Multivariate analysis | |
| | Coef (95% CI) | P-value | Coef (95% CI) | P-value | Coef (95% CI) | P-value | Coef (95%CI) | P-value | Coef (95% CI) | P-value | Coef (95% CI) | P-value |
| HIV | -0.03 (-0.08, 0.02) | 0.22 | -0.04 (-0.08, 0.01) | 0.12 | -0.02 (-0.06, 0.03) | 0.48 | -0.01 (-0.05, 0.03) | 0.56 | -0.01 (-0.06, 0.04) | 0.65 | -0.02 (-0.06, 0.02) | 0.32 |
| Age by 10 years | -0.10 (-0.14, -0.06) | <0.001 | -0.11 (-0.15, -0.06) | <0.001 | -0.09 (-0.13, -0.06) | <0.001 | -0.10 (-0.14, -0.07) | <0.001 | -0.12 (-0.15, -0.08) | <0.001 | -0.13 (-0.17, -0.09) | <0.001 |
| BMI | 0.01 (0.01, 0.02) | <0.001 | 0.02 (0.01, 0.02) | <0.001 | 0.01 (0.01, 0.02) | <0.001 | 0.01 (0.01, 0.02) | <0.001 | 0.01 (0.005, 0.02) | <0.001 | 0.01 (0.01, 0.02) | <0.001 |
| Current smoking | 0.02 (-0.21, 0.25) | 0.85 | | | 0.03 (-0.17, 0.23) | 0.79 | | | 0.04 (-0.17, 0.26) | 0.68 | | |
| Alcohol use | 0.04 (-0.09, 0.16) | 0.55 | | | 0.02 (-0.09, 0.12) | 0.77 | | | 0.02 (-0.09, 0.14) | 0.69 | | |
| HCV | -0.03 (-0.15, 0.09) | 0.62 | | | -0.01 (-0.11, 0.09) | 0.80 | | | -0.01 (-0.11, 0.10) | 0.92 | | |
| HBV | 0.07 (-0.05, 0.18) | 0.24 | | | 0.04 (-0.05, 0.12) | 0.41 | | | 0.03 (-0.07, 0.12) | 0.59 | | |
| FH of osteoporosis | -0.04 (-0.11, 0.03) | 0.27 | | | -0.07 (-0.12, -0.01) | 0.029 | -0.05 (-0.10, 0.002) | 0.059 | -0.04 (-0.10, 0.02) | 0.23 | | |
| Still menstruation | 0.09 (0.04, 0.14) | 0.001 | 0.05 (0.004, 0.10) | 0.035 | 0.05 (0.01, 0.10) | 0.018 | 0.01 (-0.03, 0.05) | 0.59 | 0.06 (0.01, 0.11) | 0.012 | 0.01 (-0.04, 0.05) | 0.79 |

*Coef = coefficient, BKK, residential location in Bangkok.

All variables with p-value <0.20 in the univariate model together with age and BMI were adjusted in the multivariate model. BMI: body mass index; HCV: hepatitis C; HBV: hepatitis B; FH: family history.

P = 0.035]. Both HIV+men and HIV+women had higher proportion of vitamin D deficiency (men: 33.5% in HIV+ vs. 13.1% in HIV-, P = 0.002; women: 37.6% in HIV+ vs. 20% in HIV-, P = 0.04). c Serum iPTH levels were also significantly higher in PWH compared to the healthy controls (Table 5).

The level of BTMs, serum P1NP and serum CTX, were significantly higher in both HIV +men and HIV+women tPWH compared to healthy controls (Table 5). Only serum P1NP remained significantly higher in HIV+men in the adjusted model (Table 6). There were no differences in the plasma levels of the inflammatory biomarkers, hs-CRP and IL-6, between men and women with and without HIV.

Being infected with HIV among males was an independent predictor for lower levels of 25 (OH)D in the adjusted model containing age, BMI and current residential location. However, serum 25(OH)D level and iPTH were not correlated with BMD results (p-values >0.2). Among PWH, the exposure to TDF or efavirenz did not affect serum 25(OH)D level. Estimated glomerular filtration rates were comparable between the participants with and without HIV for both sexes, and none of them had hypophosphatemia.

## Discussion

In the current study, we assessed the BMD and vitamin D status in virologically suppressed PWH aged over 50 years compared to healthy controls. We found that PWH had lower BMD at the total hip and femoral neck. PWH had more clinical risk factors for osteoporosis and/or fracture, such as smoking, and co-infection with either chronic hepatitis B or hepatitis C than their respective controls. In men, BMI and smoking were independent factors that affected the BMD results at all sites, while age was an independent factor affecting BMD at the femoral

**Table 5. Calciotropic hormones, and markers of bone turnover and inflammation.**

| Laboratory results | Men | | | Women | | |
|---|---|---|---|---|---|---|
| | HIV- (n = 61) | HIV + (n = 209) | P-value | HIV- (n = 41) | HIV + (n = 126) | P-value |
| Calcium (mg/dL) | 9.2 (8.9–9.5) | 8.9 (8.5–9.2) | <0.001 | 8.8 (8.3–9.3) | 8.9 (8.5–9.3) | 0.83 |
| Phosphate (mg/dL) | 2.9 (2.7–3.2) | 3.3 (2.9–3.7) | <0.001 | 3.4 (3.2–3.6) | 3.7 (3.4–4.1) | <0.001 |
| Albumin (g/dL) | 4.1 (3.9–4.3) | 4.4 (4.1–4.6) | 0.001 | 3.9 (3.7–4.1) | 4.4 (4.0–4.6) | <0.001 |
| iPTH (pg/mL) | 34.9 (29.1–46.7) | 42.0 (31.1–58.7) | 0.010 | 33.7 (25.4–42.9) | 44.8 (36.5–55.3) | <0.001 |
| Creatinine (mg/dL) | 0.93 (0.84–1.02) | 0.97 (0.84–1.13) | 0.14 | 0.72 (0.68–0.77) | 0.76 (0.70–0.84) | 0.005 |
| Serum 25(OH)D (ng/mL) | 25.5 (23.1–31.5) | 23.3 (17.9–28.8) | | 24.2 (21.5–28.7) | 21.9 (17.9–26.8) | |
| <20, n (%) | 8 (13.1) | 70 (33.5) | 0.001 | 8 (20.0) | 47 (37.6) | 0.035 |
| ≥20, n (%) | 53 (86.9) | 139 (66.5) | 0.002 | 32 (80.0) | 78 (62.4) | 0.04 |
| P1NP (ng/mL) | 41.3 (34.9–53.7) | 48.6 (37.7–58.9) | 0.016 | 49.3 (40.5–58.8) | 58.9 (46.6–79.7) | 0.008 |
| CTX (ng/mL) | 0.34 (0.24–0.41) | 0.39 (0.29–0.51) | 0.005 | 0.35 (0.31–0.44) | 0.48 (0.36–0.65) | 0.001 |
| hs-CRP (mg/L) | 1.32 (0.46–2.32) | 1.29 (0.59–2.56) | 0.68 | 1.44 (0.67–2.82) | 1.15 (0.58–2.54) | 0.61 |
| IL-6 (pg/mL) | 6.0 (3.4–10.2) | 6.3 (4.3–9.0) | 0.64 | 6.5 (4.5–12.2) | 5.9 (4.1–7.4) | 0.10 |

Data are reported as median (IQR) or number (percentage). The difference between the people with and without HIV was compared using t-test, and Mann-Whitney two-sample statistic, where appropriate. HIV-, people living without HIV; HIV+, PWH; iPTH, intact parathyroid hormone; 25(OH)D, 25-hydroxyvitamin D; P1NP, procollagen type 1 N-terminal propeptide; and CTX, C-terminal cross-linking telopeptide of type I collagen.

neck. In women, age and BMI were independent factors affecting BMD at all sites, while still menstruating was an independent factor affecting BMD at the lumbar spine. After adjusting for clinical risk factors listed above, there were no differences in BMD at any sites between PWH and people living without HIV (Tables 3 and 4).

**Table 6. Comparison of unadjusted and adjusted bone turnover markers between people living with and without HIV of both sexes.**

| Bone turnover markers | HIV status | Unadjusted | | P-value[‡] | Adjusted | | P-value |
|---|---|---|---|---|---|---|---|
| | | Mean difference | 95% CI | | Mean difference | 95% CI | |
| **Men (n = 270)** | | | | | | | |
| P1NP (ng/mL) | Uninfected | ref | - | | ref | - | |
| | Infected | 7.15 | (1.63, 12.66) | 0.011 | 7.14 | (1.52, 12.76) | 0.013 |
| CTX (ng/mL) | uninfected | ref | - | | ref | - | |
| | infected | 0.07 | (0.02, 0.12) | 0.008 | 0.04 | (-0.02, 0.10) | 0.20 |
| 25(OH)D (ng/mL) | Uninfected | ref | - | | ref | - | |
| | Infected | -3.97 | (-6.44, -1.49) | 0.002 | -3.59 | (-6.12, -1.06) | 0.006 |
| **Women (n = 167)** | | | | | | | |
| P1NP (ng/mL) | uninfected | ref | - | | ref | - | |
| | infected | 12.72 | (2.92, 22.52) | 0.011 | 10.10 | (-2.73, 22.94) | 0.122 |
| CTX (ng/mL) | uninfected | ref | - | | ref | - | |
| | infected | 0.13 | (0.04, 0.22) | 0.004 | 0.10 | (-0.02, 0.21) | 0.097 |
| 25(OH)D (ng/mL) | Uninfected | ref | - | | ref | - | |
| | Infected | -2.49 | (-4.80, -0.18) | 0.035 | -2.97 | (-6.18, 0.25) | 0.07 |

P-values were estimated by linear regression mode.
All variables with p-value <0.20 in the univariate model together with age, BMI, current smoking and alcohol use status were adjusted in the multivariate model for P1NP and CTX. All variables with p-value <0.20 in the univariate model together with age, BMI, and living in Bangkok were adjusted in the multivariate model for serum 25(OH)D. P1NP: serum procollagen type 1 N-terminal propeptide; CTX: C-terminal cross-linking telopeptide of type I collagen; 25(OH)D:25-dihydroxyvitamin D (25(OH)D).

The effects of HIV on BMD have yielded conflicting results. Our study is in line with Bolland MJ. et al. [26] and Kooij KW. et al. [27], who found that there were no significant differences in BMD between men with and without HIV after adjusting for traditional osteoporotic risk factors. In addition, a meta-analysis among PWH, mainly age below 50 years, and age- and sex matched controls showed that HIV infection was not associated with lower BMD after adjusting for traditional risks, mainly body weight [28]. This study further demonstrated that the BMD levels in well virally suppressed older PWH were comparable to people living without HIV, including older women. Interestingly, a recent study also showed that HIV status was not independently associated with volumetric BMD [29].

In contrast, several previous studies reported that PWH had a lower BMD and a higher fracture risk [30–33]. An early meta-analysis of 11 studies reported that PWH had higher risk of having low BMD (6.4 times) and osteoporosis (3.7 times) than people living without HIV [6]. Analysis using a large database from the U.S. healthcare system identified an increase in fracture prevalence among PWH compared to their controls, however, the data regarding factors associated with bone fragility were limited [34]. Several recent studies reported that there was a modest increase or even equivalent fracture risk after demographics, comorbidities, smoking, alcohol, and BMI were adjusted in the multivariate models [35, 36].

In the present study, the percentage of participants with a history of fractures, either all fractures or low-trauma fractures, was similar between PWH and healthy controls. Nearly half of the PWH underwent T-L X-ray radiographs. Of these, 6/87 (6.9%) men and 1/67 (1.5%) woman had morphometric vertebral fractures. Baseline characteristics and BMD results were comparable between those with or without vertebral fracture screening. Unfortunately, this study did not have vertebral fracture data in the control group for comparison. However, the percentage of morphometric vertebral fractures in PWH was less than what have been reported in the general population [37–40].

The underlying mechanisms for bone loss in PWH are multifactorial, including traditional, HIV-specific risk factors and ART. ART initiation, regardless of regimen, resulted in a 2–6% decrease in BMD over the first few years [21, 41, 42] and became stable or started to increase thereafter [5, 15, 43, 44]. This study did not find an association between HIV and ART (duration of HIV diagnosis, nadir or current CD4 cell count, or baseline HIV-1 RNA) as well as BMD. Although TDF and some protease inhibitors have been shown to be associated with BMD loss and may increase fracture risk [45, 46], this study did not detect this association, possibly partly due to the lower dose of TDF use. Thirty-five percent of the participants in this study used TDF at half dose or they were switched to non-TDF regimen before enrollment into the study.

Furthermore, vitamin D plays a key role in calcium and skeletal homeostasis and there is limited evidence that suggests low vitamin D levels are related to low BMD in PWH [47]. Several studies consistently demonstrated a high prevalence of vitamin D deficiency in PWH [18]. It can be related to traditional risk factors, specific ART including efavirenz and protease inhibitors, and HIV-induced chronic inflammation [18, 48, 49]. In this study, HIV infection was significantly associated with lower 25(OH)D levels in the multivariate model which was adjusted for age and BMI. However, neither serum 25(OH)D nor iPTH level was associated with BMD. The association of TDF and efavirenz as well as serum 25(OH)D levels were also not detected. In addition, this study did not observe the association of inflammatory biomarkers, hs-CRP and IL-6, and serum 25(OH)D levels among both men and women.

On the other hand, the level of bone turnover markers, serum P1NP and serum CTX, were higher in PWH compared to healthy controls. However, only serum P1NP remained significantly higher in HIV+men in the multivariate model. The absolute differences were small and the median (IQR) values for both groups were within the reference range, and therefore,

unlikely to be clinically significant. Collectively, the results from this study indicated that bone loss or fracture risk in older virologically suppressed PWH are mainly caused by the traditional risk factors as seen in the general population. Once HIV infection was controlled after ART, bone loss or fracture risk can be reversed, at least some part, over time.

This study had some limitations. First, PWH were from a research clinic and were closely monitored and regularly followed. The findings from this study may not be applicable to other older PWH from other settings, where the resources are more limited. Second, all PWH included in the study were long term virologically suppressed, so this study could not investigate whether uncontrolled HIV could have had any significant long-term impact on the bone health or not. Third, one-third of PWH used TDF at half dose, hence this study cannot conclude whether TDF was not associated with low BMD. It is possible that lower tenofovir concentrations may lower the risk of having low BMD as seen in tenofovir alafenamide treated participants. Forth, smoking, alcohol consumption, and history of fracture were self-reported and could possibly have been underestimated. Fifth, index of bone microarchitectural quality, such as trabecular bone score, was not assessed in our study. It has been shown that PWH had worse bone microarchitecture than age- and sex-matched HIV negative persons [50]. Future studies evaluating the bone quality together with BMD may improves fracture prediction in older PWH with virological suppression. Finally, this study is a cross-sectional design, while a longitudinal study would have provided more information on the BMD changes over time and the fracture incidence, as well as the factors that influence bone loss and fracture.

In conclusion, older PWH with virological suppression did not have lower BMD and fracture rates were comparable to the general population. These results indicated that effective treatment for HIV and minimization of other traditional osteoporosis risk factors can help maintain good skeletal health and prevent premature bone loss.

## Acknowledgments

The authors would like to thank the participants for volunteering in this study and HIV-NAT 207/006 study team for their contribution. The study team members are listed below.

**HIV-NAT 207/006 study team**

Lead author–Anchalee Avihingsanon (anchaleea2009@gmail.com)

**HIV-NAT, Thai Red Cross–AIDS Research Centre:** Anchalee Avihingsanon, Aroonsiri Sangarlangkarn, Sivaporn Gatechompol, Stephen J Kerr, Tanakorn Apornpong, Sasiwimol Ubolyam, Jaravee Jirapasiri, Supalak Phonphithak, Khuanruan Supakawee, and Sarapol Thongphan

**Division of Endocrinology and Metabolism, Department of Medicine, Faculty of Medicine, Chulalongkorn University:** Sarat Sunthornyothin, Lalita Wattanachanya

**Division of Nuclear medicine, Department of Radiology, Faculty of Medicine, Chulalongkorn University:** Tawatchai Chaiwatanarat

**Division of Neurology, Department of Medicine, Faculty of Medicine, Chulalongkorn University:** Aurauma Chutinet

**Division of Infectious Diseases, Department of Medicine, Faculty of Medicine, Chulalongkorn University:** Opass Putcharoen

**Division of Perioperative and Ambulatory Medicine, Department of Medicine, Faculty of Medicine, Chulalongkorn University:** Sarawut Siwamogsatham

**Division of Cardiovascular Diseases, Department of Medicine, Faculty of Medicine, Chulalongkorn University:** Sudarat Satitthummanid, Pairoj Chattranukulchai, Smonporn Boonyaratavej Songmuang and Aekarach Ariyachaipanich.

## Author Contributions

**Conceptualization:** Lalita Wattanachanya, Sarawut Siwamongsatham, Pairoj Chattranukulchai, Anchalee Avihingsanon.

**Data curation:** Lalita Wattanachanya, Anchalee Avihingsanon.

**Formal analysis:** Tanakorn Apornpong, Stephen Kerr.

**Funding acquisition:** Sarawut Siwamongsatham, Pairoj Chattranukulchai.

**Investigation:** Lalita Wattanachanya, Sarat Sunthornyothin, Hay Mar Su Lwin, Sivaporn Gatechompol, Win Min Han, Thanathip Wichiansan, Tawatchai Chaiwatanarat, Anchalee Avihingsanon.

**Methodology:** Lalita Wattanachanya, Anchalee Avihingsanon.

**Validation:** Lalita Wattanachanya, Anchalee Avihingsanon.

**Writing – original draft:** Lalita Wattanachanya.

**Writing – review & editing:** Lalita Wattanachanya, Sarat Sunthornyothin, Hay Mar Su Lwin, Sivaporn Gatechompol, Win Min Han, Thanathip Wichiansan, Sarawut Siwamongsatham, Pairoj Chattranukulchai, Tawatchai Chaiwatanarat, Anchalee Avihingsanon.

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
