## [Decision Letter · Decision Letter 0]

1 Aug 2022

PONE-D-22-09858Bone mineral density among virologically suppressed Asians older than 50 years old living with and without HIV: a cross-sectional studyPLOS ONE

Dear Dr. Avihingsanon,

Thank you for submitting your manuscript to PLOS ONE. After careful consideration, we feel that it has merit but does not fully meet PLOS ONE’s publication criteria as it currently stands. Therefore, we invite you to submit a revised version of the manuscript that addresses the points raised during the review process.

Each of the two reviewers has made specific recommendations for something to add so please address this in your revised version.

We look forward to receiving your revised manuscript.

Kind regards,

Julie AE Nelson, PhD

Academic Editor

PLOS ONE

Journal Requirements:

The author(s) received no specific funding for this work. The funders had no role in study design, data collection and analysis, decision to publish, or preparation of the manuscript.

I have read the journal's policy and the authors of this manuscript have the following competing interests: KR received honoraria or consultation fees and/or participation in a company sponsored speaker's bureau from: ViiV, Merck, Jensen-Cilag, Mylan and Gilead. The rest of the authors declare no conflict of interest.

5. One of the noted authors is a group or consortium HIV-NAT 207/006 study team. In addition to naming the author group, please list the individual authors and affiliations within this group in the acknowledgments section of your manuscript. Please also indicate clearly a lead author for this group along with a contact email address.

Reviewers' comments:

Reviewer's Responses to Questions

**Comments to the Author**

1. Is the manuscript technically sound, and do the data support the conclusions?

Reviewer #1: No

Reviewer #2: Yes

2. Has the statistical analysis been performed appropriately and rigorously? 

Reviewer #1: Yes

Reviewer #2: Yes

3. Have the authors made all data underlying the findings in their manuscript fully available?

Reviewer #1: Yes

Reviewer #2: Yes

4. Is the manuscript presented in an intelligible fashion and written in standard English?

Reviewer #1: Yes

Reviewer #2: Yes

5. Review Comments to the Author

Reviewer #1: The present manuscript is sound, however, It appears that a critical piece of data is missing. The adjusted BMD values adjust for age, BMD, and other traditional osteoporotic risk factors. The statistical analysis seemed appropriate and was rationalized. All data underlying the findings in their manuscript were made fully available. Moreover, the manuscript is presented in an intelligible fashion and written in standard English.

Reviewer #2: Introduction:

Well written overview of the current literature, as well as justification for the research conducted.

Methods:

Study population: further information should be detailed on how the cohort and sample selected for this specific AMH study were derived. Sample size (462 of which 256 were Women living with HIV (WLWHIV) ) appears adequate to make conclusions although the wide age range ( 12-50 years of age) result in small numbers in certain age groups.

Statistical Analyses appear appropriate.

Results:

The findings indicate the expected relationship between younger age and higher AMH levels. Of Interest, and in support of their hypothesis, the authors found that among the women over 35 years of age, the AMH levels were associated with HIV status and shortened telomere length in PBMCs, with WLWHIV having lower AMH levels compared to women of similar age range who were not HIV infected. These findings support earlier studies that suggested WLWHIV and provide a potential laboratory marker, AMH, to help provide guidance for WLWHIV with regards to fertility, and child spacing decisions. Some limitations to the findings include that the WLWHIV had increased risk of other risk factors that could also impact on telomere length and ovarian function such as smoking, and drug use.

Conclusions: well thought out and not overreaching.

In summary, an interesting and useful manuscript in terms of better understanding of the role of AMH over time and hypotheses generating in terms of differences seen among WLWHIV and women without HIV in terms of ovarian aging and reproductive health.

6. PLOS authors have the option to publish the peer review history of their article (what does this mean?). If published, this will include your full peer review and any attached files.

Reviewer #1: No

Reviewer #2: No

---

## [Author Response · Author response to Decision Letter 0]

30 Sep 2022

Reviewers' comments:

Reviewer #1: 

1. Improve mechanics, table organization, spacing as well as formatting of tables in the document.

RESPONSE: Thank you for the suggestion. We have managed to make sure that all the tables are well formatted and easy to read.

2. A few lines, for example, Line 57 appear to need citations.

RESPONSE: Thank you for pointing out this this. We have carefully rechecked to ensure that all the sentences had proper citations, including the sentence in line 58-59. Reference 3, 4, and 50 were added into the References section.

3. The reproductive history in Table 1 is unclear. Is menstruation exhausted referring to the number of women that have reached menopause?

RESPONSE: Thank you for. Yes, “menstruation exhausted” referred to the number of women that have reached menopause. We have added “n (%)” after each category of reproductive history in Table 1.

4. Empty boxes in tables 3 and 4

RESPONSE: All variables with p-values ≤ 0.20 from the univariate analyses were included in the multivariable regression model. Therefore, all the empty boxes in Table 3 and Table 4 were left blank because these variables that were not significant in the univariate models were not included in the multivariate analyses. The detailed methodology regarding the adjusted models was provided in the Statistical analysis section of, line 132-133. 

5. What units are being presented in table 6?

RESPONSE: We have added the units of P1NP, CTX and 25(OH)D in Table 6.

6. Were any of the women in the study on hormonal replacement therapy?

RESPONSE: All the women in the study were not on hormonal replacement therapy.

7. In the abstract it is written, “After adjusting for age, BMI, 45 and other traditional osteoporotic risk factors, BMD of virologically suppressed older PWH did not differ from participants without HIV”, however, no data is shown highlighting adjusted BMD. A similar statement is made in the discussion, lines 259-260, which states, “after adjusting for clinical risk factors listed above, there was no difference in BMD at any sites between PWH and people living without HIV”, however, adjusted BMD values are not in the manuscript. All that is shown in table 2, which has the unadjusted BMD values, which are lower in people with HIV compared to people without HIV.

RESPONSE: Analysis of the comparison of adjusted BMD between HIV+ and HIV- participants of both sexes (shown below) indicated that the BMD values after adjustment were comparable between HIV- and HIV+ of both sexes.

 Comparison of adjusted BMD between HIV+ and HIV- participants of both sexes

Site HIV status Absolute BMD (g/cm2) Adjusted model* P-valueⱡ

 Mean difference 95% CI 

Men (n=297)

Lumbar spine uninfected 0.96 (0.14) ref - 

 infected 0.94 (0.16) -0.002 (-0.04, 0.04) 0.93

Total hip uninfected 0.91 (0.13) ref - 

 infected 0.86 (0.13) -0.03 (-0.06, 0.003) 0.076

Femoral neck uninfected 0.74 (0.12) ref - 

 infected 0.71 (0.11) -0.02 (-0.04, 0.01) 0.26

Women (n=184)

Lumbar spine uninfected 0.88 (0.18) ref - 

 infected 0.85 (0.16) -0.04 (-0.08, 0.01) 0.12

Total hip uninfected 0.81 (0.17) ref - 

 infected 0.79 (0.13) -0.01 (-0.05, 0.03) 0.56

Femoral neck uninfected 0.66 (0.21) ref - 

 infected 0.64 (0.12) -0.02 (-0.06, 0.02) 0.32

 *Adjusted for age, BMI, and smoking status in men and for age, BMI, and menstruation 

 status in women; ⱡ P-values were estimated by linear regression models

• We have demonstrated in Table 3 and Table 4 that HIV status was not correlated with BMD in both men and women in the multivariate analyses. So, we did not include the table above in our manuscript. 

However, we have provided p-value “P>0.1” in the abstract as the followings:

“After adjusting for age, BMI, 45 and other traditional osteoporotic risk factors, BMD of virologically suppressed older PWH did not differ from participants without HIV (P>0.1)”. 

• In Discussion section, we have also added the table references “(Table 3 and Table 4)” in line 271-273 as the following:

“…after adjusting for clinical risk factors listed above, there was no difference in BMD at any sites between PWH and people living without HIV (Table 3 and Table 4).” 

8. Elevated circulating levels of bone turnover markers such as P1NP and CTX can be indicative of dysregulated bone turnover, which can also drive increased fracture risk. It is worth discussing what this could mean for long-term bone quality for people living with HIV. For instance, it has been shown that there is reduced bone microarchitecture in people with HIV despite comparable BMD relative to HIV age-matched sex-matched HIV negative persons, therefore even if there are no significant BMD differences there are other factors outside of BMD that may drive fracture risk.

RESPONSE: We have expanded our discussion on page 21 in Discussion section, line 335-337, as the followings:

“Fifth, index of bone microarchitectural quality, such as trabecular bone score, was not assessed in our study. It has been shown that PWH had worse bone microarchitecture than age- and sex-matched HIV negative persons [50]. Future studies evaluating the bone quality together with BMD may improves fracture prediction in older PWH with virological suppression. Finally, this study is a cross-sectional design, while a longitudinal study would have provided more information on the BMD changes over time and the fracture incidence, as well as the factors that influence bone loss and fracture.”

Reviewer #2: 

1. Ideally, a longitudinal design would have been used. The authors should comment on the rationale for only presenting cross sectional data for this cohort which is in long term follow up.

RESPONSE: Thank you for your suggestion. Currently, the longitudinal study of these participants is undergoing to follow-up on measuring the changes in BMD and the fracture incidence, as well as identifying the factors that influence bone loss and fracture.

• In the current study, we report the outcome of BMD, calciotropic hormones, vitamin D status, and bone turnover markers at the patients’ baseline visit compared to HIV-uninfected controls. 

• We have expanded our discussion on page 22 in Discussion section, line 339-341.

2. The reason that 38% of those on TDF ART were on a half dose should be further detailed. 

RESPONSE: All treatment regimens and doses were based on the physician’s decision since this is an observational study, and the reason behind it may probably due to the reduced creatinine clearance or low body weight of patients. We have added more detail in Results section (line 160-162) as the following:

“Among these, 35% used TDF at half dose. Of note, since this is an observational study, all ART regimens and doses were based on the physician’s decision and local guidelines, and the reason behind it may probably because of reduced creatinine clearance or low body weight of patients.”

---

## [Decision Letter · Decision Letter 1]

24 Oct 2022

Bone mineral density among virologically suppressed Asians older than 50 years old living with and without HIV: a cross-sectional study

PONE-D-22-09858R1

Dear Dr. Avihingsanon,

We’re pleased to inform you that your manuscript has been judged scientifically suitable for publication and will be formally accepted for publication once it meets all outstanding technical requirements.

Kind regards,

Julie AE Nelson, PhD

Academic Editor

PLOS ONE

Additional Editor Comments (optional):

Reviewers' comments:

Reviewer's Responses to Questions

**Comments to the Author**

1. If the authors have adequately addressed your comments raised in a previous round of review and you feel that this manuscript is now acceptable for publication, you may indicate that here to bypass the “Comments to the Author” section, enter your conflict of interest statement in the “Confidential to Editor” section, and submit your "Accept" recommendation.

Reviewer #1: All comments have been addressed

Reviewer #2: All comments have been addressed

2. Is the manuscript technically sound, and do the data support the conclusions?

Reviewer #1: Yes

Reviewer #2: (No Response)

3. Has the statistical analysis been performed appropriately and rigorously? 

Reviewer #1: Yes

Reviewer #2: (No Response)

4. Have the authors made all data underlying the findings in their manuscript fully available?

Reviewer #1: Yes

Reviewer #2: (No Response)

5. Is the manuscript presented in an intelligible fashion and written in standard English?

Reviewer #1: Yes

Reviewer #2: (No Response)

6. Review Comments to the Author

Reviewer #1: In addition to fully addressing all of the reviewers' comments, the authors improved the readability of the manuscript and provided further justification for the formatting.

Reviewer #2: (No Response)

7. PLOS authors have the option to publish the peer review history of their article (what does this mean?). If published, this will include your full peer review and any attached files.

Reviewer #1: No

Reviewer #2: No

---

## [Editor Report · Acceptance letter]

28 Oct 2022

PONE-D-22-09858R1 

Bone mineral density among virologically suppressed Asians older than 50 years old living with and without HIV: A cross-sectional study 

Dear Dr. Avihingsanon:

I'm pleased to inform you that your manuscript has been deemed suitable for publication in PLOS ONE. Congratulations! Your manuscript is now with our production department. 

Kind regards, 

on behalf of

Dr. Julie AE Nelson 

Academic Editor

PLOS ONE